



# 1 An interlaboratory comparison to quantify oxidative potential
# 2 measurement in aerosol particles: challenges and
# 3 recommendations for harmonisation

Pamela A. Dominutti[1], Jean-Luc Jaffrezo[1], Anouk Marsal[1], Takoua Mhadhbi[1], Rhabira Elazzouzi[1],
Camille Rak[1], Fabrizia Cavalli[2], Jean-Philippe Putaud[2], Aikaterini Bougiatioti[3], Nikolaos
Mihalopoulos[3,4], Despina Paraskevopoulou[3], Ian Mudway[5], Athanasios Nenes[6,§], Kaspar R.
Daellenbach[7], Catherine Banach[8], Steven J. Campbell[9], Hana Cigánková[10], Daniele Contini[11], Greg
Evans[12], Maria Georgopoulou[6], Manuella Ghanem[13], Drew A. Glencross[5], Maria Rachele Guascito[11,14],
Hartmut Herrmann[15], Saima Iram[16], Maja Jovanović[17], Milena Jovašević-Stojanoić[17], Markus
Kalberer[18], Ingeborg M. Kooter[19,†], Suzanne E. Paulson[8], Anil Patel[20#*], Esperanza Perdrix[21], Maria
Chiara Pietrogrande[22], Pavel Mikuška[10], Jean-Jacques Sauvain[23], Aikaterina Seitanidi[6], Pourya
Shahpoury[24], Eduardo J. dos S. Souza[15], Sarah Steimer[20#], Svetlana Stevanovic[16], Guillaume Suarez[23],
P. S. Ganesh Subramanian[25], Battist Utinger[18], Marloes F. van Os[19], Vishal Verma[25], Xing Wang[12],
Rodney J. Weber[26], Yuhan Yang[26], Xavier Querol[27], Gerard Hoek[28], Roy M. Harrison[29+], and Gaëlle
Uzu[1*].
[1] University Grenoble Alpes, CNRS, INRAE, IRD, INP-G, IGE (UMR 5001), 38000 Grenoble, France
[2] European Commission, Joint Research Centre (JRC), Ispra, Italy
[3] Institute for Environmental Research and Sustainable Development, National Observatory of Athens, Lofos
Koufou, P. Penteli, Athens, 15236, Greece
[4] Environmental Chemical Processes Laboratory, Department of Chemistry, University of Crete, Heraklion,
71003, Greece.
[5] MRC Centre for Environment and Health, and the National Institute of Health Research, Health Protection
Research Unit in Environmental Exposures and Health, Imperial College London, London, UK
[6] Center for the Study of Air Quality and Climate Change, Institute of Chemical Engineering Sciences, Foundation
for Research and Technology Hellas, Patras, Greece 26504
[7] Laboratory of Atmospheric Chemistry, Paul Scherrer Institute, 5232 Villigen PSI, Switzerland
[8] Department of Atmospheric and Oceanic Sciences, University of California at Los Angeles, 520 Portola Plaza,
Los Angeles, California, 90095, United States
[9] MRC Centre for Environment and Health, Environmental Research Group, Imperial College London, 86 Wood
Lane, London W12 0BZ, UK.
[10] Department of Environmental Analytical Chemistry, Institute of Analytical Chemistry, Czech Academy
of Sciences, Veveří 97, 60200 Brno, Czech Republic
[11] Institute of Atmospheric Sciences and Climate, ISAC-CNR, Str. Prv. Lecce-Monteroni km 1.2, 73100 Lecce, Italy
[12] Southern Ontario Centre for Atmospheric Aerosol Research, University of Toronto, Toronto, M5S 3E5, Canada





[13] Department of Pollutant Metrology, Institut National de Recherche et de Sécurité (INRS), 54500 Vandœuvre-
lès-Nancy, France
[14] Department of Environmental and Biological Sciences and Technologies (DISTEBA), University of
Salento, Lecce 73100, Italy
[15] Atmospheric Chemistry Department (ACD), Leibniz Institute for Tropospheric Research (TROPOS),
Permoserstraße 15, 04318 Leipzig, Germany
[16] School of Engineering, Deakin University, VIC 3216, Australia
[17] Vinča Institute of Nuclear Sciences – National Institute of the Republic of Serbia, University of Belgrade,
11 351 Belgrade, Serbia
[18] Department of Environmental Sciences, University of Basel, 4056 Basel, Switzerland
[19] TNO Environmental Modelling, Sensing and Analysis, Princetonlaan 6-8, 3584 CB Utrecht, Netherlands
[20] Department of Environmental Science, Stockholm University, Stockholm, 11418, Sweden
[21] IMT Nord Europe, Institut Mines-Télécom, Univ. Lille, Centre for Energy and Environment, F-59000 Lille,
France
[22] Department of Chemical, Pharmaceutical and Agricultural Sciences, University of Ferrara, Via Fossato
di Mortara 17/19, 44121 Ferrara, Italy
[23] Center for Primary Care and Public Health (Unisanté), Department of Occupational and Environment Health
(DSTE), University of Lausanne, Switzerland
[24] Environmental and Life Sciences, Trent University, Peterborough, Canada
[25] Department of Civil and Environmental Engineering, University of Illinois at Urbana-Champaign, 205 North
Mathews Avenue, Urbana, IL, 61801, United States
[26] School of Earth and Atmospheric Sciences, Georgia Institute of Technology, Atlanta, Georgia 30332,
United States
[27] Institute of Environmental Assessment and Water Research (IDAEA-CSIC), 08034 Barcelona, Spain
[28] Institute for Risk Assessment Sciences, Utrecht University, Utrecht, 3584CM, the Netherlands
[29] Division of Environmental Health and Risk Management, School of Geography Earth and Environmental
Sciences, Edgbaston, Birmingham, UK, B15 2TT, United Kingdom
† Deceased
[+] Also at: Department of Environmental Sciences, King Abdulaziz University, Jeddah, Saudi Arabia
[*] Now at: Department of Atmospheric and Oceanic Sciences, University of California at Los Angeles, Los Angeles,
CA 90095-1565, USA
[#] Also at: Bolin Centre for Climate Research, Stockholm, 11418, Sweden
[§] Also at: Laboratory of Atmospheric Processes and their Impacts, Institute of Environmental Engineering, Ecole
Polytechnique Federale de Lausanne, Lausanne, Switzerland, 1015
*Corresponding author: gaelle.uzu@ird.fr



**Abstract**

This paper presents the findings from a collaborative interlaboratory comparison exercise designed to assess oxidative potential (OP) measurements conducted by 20 laboratories worldwide. This study represents an innovative effort as the first exercise specifically aimed at harmonising this type of OP assay, setting a new benchmark in the field.

Over the last decade, there has been a noticeable increase in OP studies, with numerous research groups investigating the effects of exposure to air pollution particles through the evaluation of OP levels. However, the absence of standardised methods for OP measurements has resulted in variability in results across different groups, rendering meaningful comparisons challenging. To address this issue, this study engages in an international effort to compare OP measurements using a simplified method (with a dithiothreitol (DTT) assay).

Here, we quantify the OP in liquid samples to focus on the protocol measurement itself, while future ILCs should aim to assess the full-chain process, including the sample extraction. We analyse the similarities and discrepancies observed in the results, identifying the critical parameters (such as the instrument used, the use of a simplified protocol, the delivery and analysis time) that could influence OP measurements, and provide recommendations for future studies and interlaboratory comparisons. Even if other crucial aspects, such as sampling PM methods, sample storage, extraction methods and conditions, and the evaluation of other OP assays, still need to be standardised. This collaborative approach enhances the robustness of the OP-DTT assay and paves the way for future studies to build on a unified framework. This pioneering work concludes that interlaboratory comparisons provide essential insights into the OP metric and are crucial to move toward the harmonisation of OP measurements.

## 1. Introduction

Over the last decade, many studies demonstrated associations between exposure to ambient air pollution and adverse human health outcomes (Hart et al., 2015; Laden et al., 2006; Lepeule et al., 2012; WHO, 2021b, a). Adverse health effects attributable to particle matter (PM) are complex and diverse. Among environmental factors, PM is considered to be the largest contributor to morbidity and mortality globally (WHO, 2017). The casual mechanisms underpinning these adverse associations are diverse, with oxidative stress and inflammation, genomic alterations, damage to the nervous system function, and epigenetic alterations, among others, all cited as potential contributing pathways (Huang et al., 2022; Nicholson et al., 2022; Wilker et al., 2023; Zare Sakhvidi et al., 2022). Across these broad domains, the capacity of particles and particle-derived chemicals to cause damaging biological oxidations appears to be a unifying mechanism, both through the introduction of pro-oxidants and stable free radicals into the body, but also through secondary radical/oxidant generation through altered metabolism and induction of inflammation (Li et al., 2008). By definition, oxidative



stress is a condition where excess production of reactive oxygen species (ROS) and nitrogen species (RNS)
overwhelm endogenous antioxidant defences (Shankar and Mehendale, 2014). Generally, ROS/RNS production
in the cells is regulated within physiological limits, through the actions of antioxidant enzymes (e.g. superoxide
dismutase, catalase, etc), low molecular weight water-soluble (e.g. glutathione, ascorbate) and fat-soluble
(vitamin E) antioxidants (Alkoussa et al., 2020). This antioxidant system plays a valuable key role by limiting
ROS/RNS damage, which is associated with cytotoxicity and the induction of inflammation due to changes in
the cellular redox balance (Cassee et al., 2013; Gao et al., 2020; Kelly and Fussell, 2017; Sies, 2018). The capacity
of PM to invoke biological oxidations has therefore been proposed as a proxy measure of their toxicity, and has
been referred to as their oxidative potential (OP); either that intrinsic to their possession of pro-oxidants, or
encompassing their capacity more wholistically to simulate ROS/RNS through interaction with cells (Ayres et
al., 2008; Bates et al., 2015; Cho et al., 2005; Sauvain et al., 2008; Uzu et al., 2011).
Consequently, the OP of PM is increasingly being studied as a potentially health-relevant metric to evaluate
effects due to exposure to PM, in addition to PM mass concentration, in multiple regions across the globe (Bates
et al., 2019; Bhattu et al., 2024; Daellenbach et al., 2017; Weichenthal et al., 2019). OP is a relatively simple
estimation of PM redox activity that reflects a complex interplay of all physico-chemical properties (chemical
composition, surface-area, solubility and particle size) contributing to the ROS/RNS generation and the
oxidation of target biomolecules, or probes. Implicit within this approach is the contention that not all
constituents of ambient PM are equally as harmful, and that those that drive damaging redox reactions, either
directly, or indirectly present a greater hazard. Thus, PM composition should be considered a factor more
directly linked to adverse health effects than PM mass concentration, highlighting the need to study additional
health-relevant metrics such as the OP (Park et al., 2018).
In last decades, there has been an increased interest in measuring and developing OP studies, applying different
in vivo or in vitro assays and aerosol characterisation techniques to estimate the main sources of OP related to
PM (Guascito et al., 2023), and attempting to integrate this metric into health studies. Several acellular chemical
methods have been applied for the estimation of the OP of atmospheric particles since these assays allow faster
measurement and are less labour-intensive than cell culture or in vivo methods (Bates et al., 2019). In addition,
these assays aim to mimic the interaction between PM and different lung antioxidants (e.g. glutathione,
ascorbate..., etc) during inhalation. The acellular assays which are most commonly applied include several
probe approaches based on antioxidants or surrogates, such as the dithiothreitol assay (DTT), ascorbic acid
assay (AA), glutathione assay (GSH), Ferric-Xylenol Orange assay (FOX), 9,10-bis (phenylethynyl) anthracene-
nitroxide (BPEAnit) ROS assay and 2,7-dichlorofluorescein assay (DCFH) for bound-particles. Molecular probes
display variable sensitivities to PM components due to their unique redox potentials and chemical reaction
routes, contributing to aerosol OP values. Therefore, it may be necessary to use several assays simultaneously
for a broader assessment of the chemical species in PM potentially triggering oxidative stress and to evaluate
which of these probes might be most indicative and closely linked to health effects. Furthermore, one of the





main challenges within this rapidly expanding research field is the diversity of analytical methods and protocols
used for OP each assay, which require standardised protocols to support synthesis across the evidence base
(Ayres et al., 2008).
In 2008, a previous workshop gathered experts on OP and developed consensus statements addressing the
importance of standardised samples, the comparison between oxidative potential tests, the formulation of
consistent standard test protocols and the establishment of connections between OP tests and epidemiological
findings as reliable predictors of adverse health outcome (Ayres et al., 2008). Despite more than 15 years having
elapsed since that workshop, whilst protocols have matured, evolved, and proliferated worldwide, little
concrete work has been performed regarding the harmonization and standardization of these methods.
One of the main objectives of the RI-URBANS European project (https://riurbans.eu/) is to bring accessible
service tools to enhance air quality monitoring networks, including evaluating air pollution exposure. As OP
has been proposed and recommended as a parameter to be measured in the proposal for a new European Air
Quality Directive (Council of the European Union, 2024), an international OP interlaboratory comparison (ILC)
was launched to assess the consistency of OP measurements between participants that apply different OP DTT
protocols, hindering comparison of results obtained worldwide. The main goal of the ILC was to identify
potential discrepancies in results (obtained with the OP DTT assay, one of the most common acellular assays
used for measuring the OP) that may arise due to differences in experimental procedures, equipment, or
analytical techniques. This ILC constitutes a first step to identify potential sources of variability, resulting in the
enhancement of the overall accuracy, reliability and comparability of OP measurements.
This paper presents the setup and results of this first large ILC study based on the dithiothreitol (DTT) assay,
with a large number of participants (20 groups). The first section includes a description of how a simplified
protocol was obtained, along with the coordination/management of the ILC. Subsequently, the results are
presented in the second section, combined with statistical analysis, both for the harmonized protocol and the
"home" protocols of each participant. Finally, some major findings and recommendations are presented.

## 2. Intercomparison strategy

This ILC was proposed within the RI-URBANS European Project framework to evaluate the discrepancies and
commonalities of OP measurements obtained by the different participating laboratories. The setup of the
protocol was led by a working group of laboratories with considerable experience in oxidative potential: FORTH
(The Foundation for Research and Technology – Hellas (FORTH, Greece), NOA (National Observatory of Athens,
Greece), ICL (Imperial College of London, United Kingdom), IGE (Institute of Environmental Geosciences,
France) and UoB (University of Birmingham, United Kingdom) (i.e. the "core group"). Considerations regarding
sampling techniques of PM filters or monitoring strategic approaches are beyond the scope of this exercise.
Multiple OP assays are available; however, following a literature review, it was decided to prioritise the DTT
assay for this first ILC due to its widespread adoption and long-term application facilitating broad participation





from laboratories. The core group first produced a harmonised and simplified method, detailed in a
standardised operation procedure (SOP), that was integrated, implemented and tested by IGE, the organiser
for this ILC. This SOP is called the "RI-URBANS DTT SOP" in the following and is presented in detail in Section
2.2. Section 2.1 presents the selection procedure for some parameters of the DTT assay according to variations
observed in the literature. Sections 2.3 to 2.7 comprise different parts of the implementation of the ILC, along
with the procedure for data processing.

### 183 2.1 Testing the parameters for implementing the simplified RI-URBANS protocol

The simplified RI-URBANS DTT SOP was adapted from the original DTT protocols published in the early 2000s
( SOP1: Li et al., 2003, 2009; SOP2: Cho et al., 2005; SOP3: Kumagai et al., 2002; called SOP1, SOP2 and SOP3,
hereafter). The principle of the DTT assay relies on the production of superoxide radicals, with DTT acting as a
surrogate for cellular reducing agents. This probe contains thiol groups similar to GSH and subjects to oxidation,
forming stable cyclic disulphides by donating electrons to oxygen through intermediate redox-active species
from PM. In the assay proposed by Kumagai et al. (2002), PM is incubated with DTT and its rate of oxidation
over time is assessed through the use of 5,5'-dithiobis-(2-nitrobenzoic acid) (DTNB) with the reaction product
2-nitro-5-thiobenzoic acid (TNB), being detected at 412nm. Whilst DTT is in excess, the rate of DTT oxidation
is proportional to the concentration of redox-active species in the PM (Cho et al., 2005; Sauvain et al., 2008).
The DTT loss over time can then be expressed per unit concentration of PM (usually µg) or by volume of air
($m^3$) to provide a measure of the intrinsic reactivity of the particles and assess human exposure (Cho et al.,
2005), respectively.
A review of the pioneer DTT assay protocols revealed differences, and therefore to derive a simplified protocol,
some variations of the parameters were examined by the ILC organiser laboratory. These results were
evaluated by the RI-URBANS core group to obtain the final harmonized protocol.
Three key parameters were tested. First, the necessity for the inclusion of trichloroacetic acid (TCA) was
evaluated. In the original DTT protocol by Kumagai et al. (2002), the reaction between samples and DTT is
quenched at a specified incubation time by the addition of 1.0 ml of 10% TCA to the incubation mixture.
Subsequently, 0.5 ml of the reaction mixture is extracted and combined with DTNB and tris–HCl buffer (pH 8.9).
However, more recently, Li et al. (2009) demonstrated that DTNB rapidly reacts with DTT, with the absorption
reaching its maximum value immediately and remaining stable for over two hours. In our initial tests, we found
that this parameter was consistent showing no differences over all the samples tested with or without TCA
(Figure S1). Thus, in the simplified OP RI-URBANS SOP, TCA addition was omitted, and we introduced the DTNB
solution directly into the mixture at the prescribed incubation times and recorded the absorption after the 30-
minute reaction.



Second, EDTA is present in some OP DTT protocols, whether in the buffer of the incubating PM sample with
DTT (Kumagai et al., 2002; Li et al., 2009), or in the titration mixture of DTNB (Cho et al., 2005; Kumagai et al.,
2002). EDTA is a strong chelating agent which is widely used in biological assays to prevent microbial
contamination and to facilitate cell lysis and the extraction of cellular components. Using EDTA is helpful in
purifying buffers at a low cost and decreasing a high rate of DTT loss in the blank by scavenging metal ions.
However, it can lead to artefacts during the assay. This is especially critical when used in the reaction mixture
with the PM, where it could induce complexation with redox changes. Moreover, this is particularly relevant for
iron where complexation increases solubility and the resultant EDTA-Fe complex is redox-active (Gao et al.,
2024), or in the opposite, EDTA can chelate some metallic species, preventing their reactivity (Charrier and
Anastasio, 2012). In addition, EDTA also has antioxidant properties, which may compete in the solution with
DTT (Thbayh et al., 2023). Our results on different samples show that the presence of EDTA in the buffer leads
to underestimation of the OP DTT loss rates in comparison to without EDTA. An impact was observed in the
solutions, mainly the copper reference solution (1 μM) and an ambient PM filter for testing both protocols
(Figure S2). That augmentation could also be related to an increase in the blank absorbance without EDTA due
to the impurities in the buffer, but this is controlled by subtracting the blank. Little or no impact was shown on
the 1,4-naphthoquinone solutions, an organic component. To prevent such undesirable interactions, many
laboratories have introduced Chelex® 100, a sodium-form resin to purify the buffers used in the OP DTT assay
(Calas et al., 2017, 2018; Charrier et al., 2015; Charrier and Anastasio, 2012), or used commercially available
high-purity (e.g. LC-MS/MS grade) water and buffer mixture to create the OP reaction medium (Shahpoury et
al., 2019, 2022). Nevertheless, the Chelex® commercial resin comes with a basic pH and requires a pre-
treatment that could be an extra source of error for this current first international intercomparison. Finally,
EDTA was not included in the SOP to prevent complex ligand chemistries, and because the use of Chelex® was
complex to introduce for such first ICL; high-grade buffer powders were sent to all the participants instead.
Third, original protocols include the use of a Tris-HCl buffer at pH 8.9 in the solution of titration with DTNB,
and this remains widely used. However, Li et al. (2009) showed that the pH of the solution drives both the
catalytic redox reaction rate and also the molar extinction coefficient of the product of the reaction, TNB. This
report confirmed the previously obtained results by Danehy et al. (1971) that showed that DTNB suffers from
alkaline decomposition above pH 8, increasing absorbance values (Figure S3). As a result, for the simplified RI-
URBANS DTT SOP, the core group selected a potassium phosphate buffer at a physiological pH of 7.4 to replace
tris-HCl in the DTNB solution. This prevents the DTNB alkaline decomposition because the pH is in the
favourable range of 5.5-8 where TNB is in the $TNB^{2-}$ form and the DTT+DTNB system does not show significant
pH-dependant changes in the absorbance values (Li et al., 2009).
Finally, the simplified RI-URBANS DTT SOP also includes a variation considering the instrument used for the
measurements. The SOP was elaborated and tested for both cuvette and plate reader spectrophotometers,





including the potential application of automatic samplers. However, the simplified SOP remains to be tested for
other instruments (such as Liquid waveguide capillary cell instrument (LWCC)).

**2.2 Simplified RI-URBANS DTT SOP and other measurements**
The simplified RI-URBANS DTT SOP is proposed in SI-1. It contains a first step for the preparation of the
reagents needed for the analyses included in the ILC. A second step describes how to perform a calibration of
the analytical device, using a DTT calibration curve with at least 4 points for a concentration range between 0
and 60 μM (titration with 1mM DTNB and reading of TNB formation at 412 nm). In the third step, the SOP
defines the performance of the measurements for the test samples provided, including the assessment in
triplicates of each test sample and control points (blanks). It should be noted that the DTT protocol also
integrates the analysis of control points (blanks), which allows quantifying the inherent DTT background
oxidation.
The duration of the analyses required for the ILC is variable depending on the instrument used. About 30 min
completion time is required, including the assessment of ILC test samples and control points, when performed
with a plate reader, and a similar time is needed to perform the calibration curve. When a cuvette-type
spectrometer is used, the total analysis time can be at least 2 hours to perform all the triplicates and the
corresponding calibration curve.
Furthermore, apart from the analytical equipment for monitoring the chemical reactions, the simplified
protocol also requires some standard laboratory equipment and conditions, such as access to ultrapure water
(18.2 Mohm cm$^{-1}$, TOC <5 ppb) for the preparation of reagents, the use of vortex for the homogenisation of
samples, refrigerated baths to conserve the reagents and transparent 96-wells plates and dark tubes. The
samples also need to be kept under agitation during the experiment and at a constant temperature of 37.4°C.
The calculation of the chemical reaction rate of OP DTT during this ILC involves a conversion using a calibration
curve. Once the results are obtained, the kinetics of the DTT oxidation can be calculated by subtracting both the
intrinsic absorption of each sample (absorption obtained from the samples before the addition of reagents to
remove a potential matrix effect between samples) and the inherent DTT auto-oxidation rate (slope of a Control
sample) from the DTT consumption rate in the presence of PM.
Participants were asked to perform additional OP measurements on the same samples, following the protocol
in use in their laboratory ("home protocol(s)") if they wanted to do so. In this case, they had to provide the
results in the particular units requested, depending on the applied assay.


### 2.3 Type of samples – test materials

The ILC was performed using three samples (SP1-SP3), including different concentrations of ambient PM and commercial positive control (1,4-naphthoquinone, CAS [130-15-4], Sigma Aldrich), all extracted in ultra-pure water and prepared by the ILC organiser. Providing ultra-pure water extracts - instead of filter fragments - to the participants was selected for the current first ILC to avoid additional uncertainties associated with the use of different procedures of sample extraction, different quality of the ultra-pure water for sample extractions, and changes linked to the processing equipment available. More specifically, the samples sent to the participants included SP1: 1,4-naphthoquinone solution (reference compound, 5 µg mL$^{-1}$), SP2: extract from a PM sample influenced by biomass burning emission (obtained from a chamber experiment, at 25 µg mL$^{-1}$), and SP3: an urban PM extract highly influenced by traffic emission (from pooled roadside samples obtained from TEOM - FDMS reference samplers, at 25 µg mL$^{-1}$). Additionally, a fourth sample, SP4: a sample extracted from a blank/clean quartz filter was sent to the participants, but it was not included in the evaluation since the measured values were close to the instrument limits of detection for most participants. For each of the 4 samples, all the sub-samples, distributed to participants, resulted from a unique 1L solution obtained from the original sample substrate. For instance, SP1 and SP3 were powders that were solubilised and homogenised for 75 min by vortex agitation in ultra-pure water. SP2 and SP4 were quartz fibre filters subjected to a 75-minute vortex extraction in ultrapure water.

Several 5 ml sample aliquots in dark polypropylene tubes were sent to each participant, according to their needs, allowing them to perform triplicate measurements for the RI-URBANS DTT SOP and all the "home" OP protocols implemented in their labs.

Solid potassium dihydrogen phosphate (CAS [7778-77-0], Roth), di-potassium hydrogen phosphate (CAS [7758-11-4], Roth), 5,5'-Dithio-bis-(2-nitrobenzoic acid, CAS [69-78-3], Roth) and 1,4-Dithiothreitol (CAS [3483-12-3], Roth) were also distributed to the participants, to prepare the solutions for the RI-URBANS DTT SOP, including the DTT solution, the Dithio-bis-(2-nitrobenzoic acid) (DTNB) solution, and the potassium phosphate buffer solution.

### 2.4 Transport of samples, ILC performance and duration

Test samples (SP1-SP4) were shipped to all participants on 17th January 2023 via courier in refrigerated and isolated ice packs, and received as chilled liquid samples. The parcels were delivered between 18th January and 2nd February 2023. On average, the participants received the samples in 3±2.5 days and performed the analysis in 14±10.5 days and up to 31 days after reception of samples. The recording of these parameters allows their integration to the multiple linear models used in this work.

### 2.5 Reporting of the results

Participants were asked to report OP DTT results from the RI-URBANS SOP in nmol DTT min$^{-1}$ µg$^{-1}$ and the % DTT consumed in µg$^{-1}$ min$^{-1}$, applying three decimal digits for all three replicates of test samples. An Excel spreadsheet with all the calculations pre-included was prepared by the ILC organiser and shared with all the





participants to avoid calculation errors and to facilitate the standardisation of results. In addition, participants
were invited to report, under the same format, the values for other OP tests, such as OP DTT "home", and other
OP tests like AA, DCFH, OH, ESPR, GSH and RP (routinely applied by each participant) on the same samples.

### 2.6 Number of participants

It is worth noting that for the first time, a total of 20 research groups participated in the exercise: 14 of them
from Europe, including the United Kingdom, Italy, France, Switzerland, Greece, Germany, Serbia, the Czech
Republic, the Netherlands and Sweden, 3 participants from the United States, 2 participants from Canada and
1 from Australia. The participants were invited for their contribution to the RI-URBANS project, through a
public call to participate, or because they contacted the ILC organiser directly and were selected, due to their
active role in the OP scientific community. The ILC was performed using anonymous participation; thus, a
number was randomly assigned to each participant to present the results. Participant L5 cancelled his
participation in the ILC and two participants (L3 and L16) did not send their results for RI-URBANS DTT SOP.

### 2.7 Data evaluation

The ILC results were analysed by the European Commission Joint Research Centre (JRC), which provides
independent, evidence-based science and knowledge support for EU policies and in conducting ILC exercises.
The participation of an external independent evaluation was required following the International Global
Standard ISO 5725-2, related to the accuracy of measurement methods and results (Part 2: Basic method for
the determination of repeatability and reproducibility of a standard measurement method). The data
assessment includes statistical evaluation of homogeneity, ageing, and repeatability, as well as the
establishment of the assigned values and the related methods.

### 2.7.1 Estimation of the assigned value and participants' performance

Different methods can be applied to determine the assigned (consensus) value for a comparison exercise when
no reference or certified reference material is available. The first one would be to use a robust mean and
standard deviation including all participants, but this could become statistically ineffective if the number of
participants is below twenty. Further, the results we obtained were not normally distributed (Figure S4),
compromising the accuracy of the robust mean of the samples tested. Thus, three different alternative
approaches were evaluated in this ILC until meeting the requirements of Standard ISO 5725-2. The first one is
based solely on the results obtained by the ILC organiser, where the samples were prepared, and the
homogenisation tests were performed. The assigned values obtained from this approach include a coefficient
of variability of 11%, 7% and 6% for SP1, SP2 and SP3, respectively. The second approach applies the Q/Hampel
test, a robust algorithm useful for data sets with outliers to calculate the robust standard deviation and mean
of the data set. This method is largely applied to the statistical analysis of interlaboratory studies. The
Q/Hampel test results (integrating the average of the results from all the participants) present COVs of 60%,
61% and 70% for SP1, SP2 and SP3, respectively. However, the values obtained for the dataset were deemed



too high to guarantee the good performance of the Q/Hampel test calculation. The third approach relies on
calculating the assigned (consensus) value from the average and standard deviation for a specific reference
group, including 6 participants selected (without knowing their results) due to their previous pioneering
experience in developing OP protocols and measurements, taken as their strong expertise in the domain (>10
publications in the field). The participants selected were the L2, L4, L8, L12, L13 and L19. This approach leads
to coefficients of variability of 31%, 33%, and 47% for SP1, SP2 and SP3, respectively. Since the first two
methods showed a contrasted variability, one with too little and the other with too much variability, the
assigned values were calculated using the third approach, integrating the average and standard deviation of the
results from this selected experienced OP participants.
The individual performance of each group was further evaluated using z-scores, a metric indicating the
deviation of each data point from the assigned value as compared with the standard deviation for proficiency
assessment $\sigma^*$. For each laboratory and test sample, the z-score is computed using the formula: *z = ($x_i$-X)/ $\sigma^*$*,
where $x_i$ is the result from participant *i*, X is the sample assigned value (0.53, 0.14, 0.07 nmol min$^{-1}$ µg$^{-1}$), and $\sigma^*$
is calculated as the reproducibility standard deviation among the six selected participants (0.16, 0.06, 0.04 nmol
min$^{-1}$ µg$^{-1}$). An "action signal" is triggered if a participant's entry produces a z-score exceeding +3 z or falling
below -3 z, indicating a deviation of more than 3 standard deviations from the assigned value. Similarly, a
"warning signal" is raised for a participant z-score above +2 z or below -2 z, representing a deviation between
2 and 3 standard deviations. A participant z-score between -2 z and +2 z signifies satisfactory performance
concerning the standard deviation for proficiency assessment.

### 363 2.7.2 Analysis of the variability of the results

Additionally, the statistical distribution of results was evaluated using multiple linear regression models. In the
first model, the effects of the protocol variables on the measured OP were investigated using linear regression
models adjusted on the instrument (3-class variables: plate reader, cuvette and LWCC), delivery time (time
between sample shipment and reception, continuous), and analysis time (time between reception and analysis,
continuous). An additional model (M2) compared the RI-URBANS SOP and the DTT-home protocols was further
adjusted based on the protocol (2-class variable: RI-Urbans, DTT-home). Finally, the evaluation of the average
performances (3-class variables: low - 0<|z-score |<2, middle - 2<|z-score |<3, and high - |z-score |>3) was
added in the M2 model, to assess whether performance affected DTT activity in the same direction (i.e. positive
or negative), while considering other protocol variables. Each model was run separately for SP1, SP2 and SP3
samples. All analyses were conducted using R (version 4.2).





## 3. Results and Discussion

Out of the group with a total of 20 participants, 18 presented results obtained using the RI-URBANS DTT SOP. Different instruments were used to apply the simplified SOP. Overall, 9 participants (47.5%) used the cuvette-type spectrophotometer, 8 (42%) used the plate reader-type spectrometer, and 2 (10.5%) implemented LWCC measurements (one participant used two instruments).

### 3.1 Homogeneity of the samples

An initial assessment of the homogeneity of the OP measurements with the 3 test samples was performed by the ILC organiser, using a plate-reader type protocol, deriving the mean and the standard deviation of 10 replicate analyses performed on the same day. The results obtained from the coefficient of variations, showed the sample variabilities were up to 12%, 7%, and 9% for SP1, SP2 and SP3, respectively (Figure S5), showing a higher variability for the 1,4-naphthoquinone solution (SP1) compared to the two filter extracts (SP2 and SP3).

The overall uncertainty of the OP DTT assay has been evaluated to 18% for $PM_{10}$ and 16.3% for $PM_{2.5}$ by Molina et al., 2020. Despite some differences observed between our samples, the results are deemed acceptable, presenting a variability of around 10%, which indicates a very good performance of the analysis.

### 3.2 Ageing of samples

To reduce the number of parameters affecting the preparation of the sample solutions liquid solutions of each sample were prepared and sent to the different participants. However, liquid samples can undergo ageing processes, impacting OP levels over time. For this purpose, an ageing test was performed by the ILC organiser to evaluate the potential changes over time. It consisted of regularly implementing the RI-URBANS DTT SOP to obtain the values of each test sample over time. Figure S6 shows these results, where SP2 and SP3 do not show a strong change over time, while sample SP1 presents a pronounced ageing effect. In routine tests of ICL organizer, the 1,4-naphthoquinone mother solution at such high concentration is usually stable in a glass container for weeks but here, potential interaction with the PP tubes' inner surface may have happened. Consequently, ageing could be a variable of importance for the participants who analysed the samples toward the end of the required period, and such parameter (date of analysis) was thus included in the parameters to be tested for the research of critical parameters.

### 3.3 Statistical distribution of results: Participants' variability

In order to assess the intra-laboratory variability of the results, the coefficient of variation (COV = standard deviation/ mean * 100) of the results for each sample and each laboratory are presented in Figure 1, while the standard deviations of the replicates reported for each sample are presented in Figure S7 and table S1. Overall, higher COVs are observed for SP3 and SP1, where most participants (44.4 and 38.9%, respectively) presented higher values for this sample compared to the SP2 sample. Specifically, high average COV values are observed





by L1, L12, L14, L15 and L21, with an average variation higher than 40%. Only a few participants (6 groups)
presented a variation lower than 10% for the three samples. This is the same pattern for the results obtained
by the ILC organiser during the homogeneity test (see 3.1.), where the COV for SP1 was larger than those for
SP2 and SP3 but with the highest COV below 15%. These findings confirm more homogenous results for
samples SP2 and SP3 compared to SP1 but could also indicate that some participants failed to achieve
repeatability observed by the ILC organiser. However, some groups (L2, L10, L19, L20) were able to produce
very homogeneous results with COV < 10% for all 3 samples.

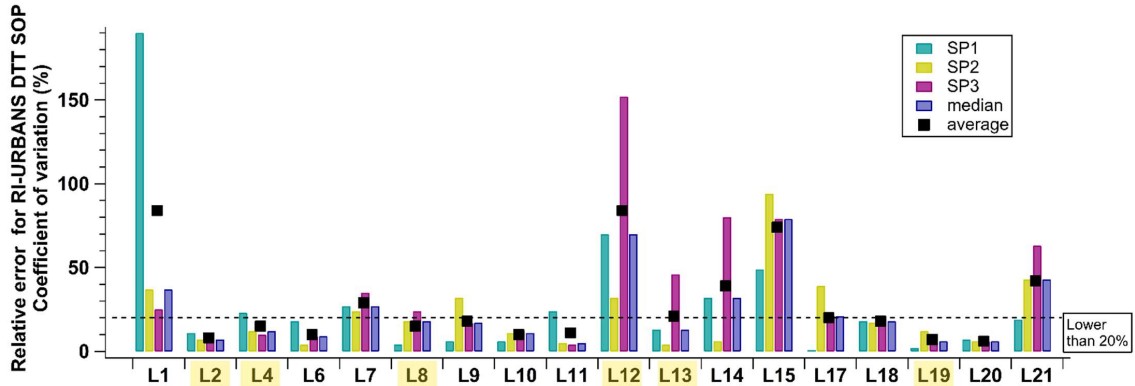


**Figure 1: Coefficients of variation of each participant (L1 to L21) for the three samples in triplicates**
**tested using the RI-URBANS DTT protocol and the median and mean repeatability for each laboratory.**
**Yellow-highlighted participants are the ones selected for the calculation of the assigned values. The**
**dashed line indicates the participant with COV lower than 20%.**

**3.4 Laboratory performances**
The assessment of laboratory performances first presents the bias in results across participant groups
compared to the assigned values and their associated standard deviation for each sample. As illustrated in
Figure 2, SP1 exhibited the highest variances, ranging from 130% to -35%, with only five groups displaying
differences within ±10%. The distribution of results for SP1 indicated a mix of overestimations and
underestimations. For SP2, differences are within a narrower range from 43% to -7%, primarily favouring
overestimations. For this sample, 12 participants returned results that were within ±10% of the assigned value
(see highlighted laboratory numbers in Figure 2). Finally, the results for SP3 demonstrated the least variation
among participants, with differences ranging from 30% to -6%, and 16 participants within ±10% of the
assigned value, again favouring overestimation compared to the assigned value. In total, 14 laboratories
obtained data with ±10% difference to the assigned value for SP2 and SP3 (see highlighted laboratory numbers
in Figure 2). These results show again that the reference samples with 1,4-naphthoquinone (SP1) most





probably present some characteristics leading to this variability and may be associated with a less stable
solution or led to saturation of some detectors regarding the relatively high concentrations, while the samples
from filter extractions do not. Additionally, it is interesting to note that there is apparently no systematic
pattern where a given participant would obtain out-of-range results for all samples. The results are really
diverse, and most participants can obtain "acceptable" results for one or two samples (SP) and larger variability
associated with one "unacceptable result" for one of them. In 2020, Molina et al. (2020) explored the total
uncertainty of OP DTT of a collection of samples and evaluated it to 18% for PM10 and 16.3% for PM2.5. The
leading factors identified were the DTT consumption rate (regression and repeatability of experimental data)
and the extraction volume operations (pipette). This underscores the need for further investigations on the
experimental causes of the variations observed, possibly in the next ICL.





**Figure 2. Percentage differences from the assigned (consensus) value for each sample (SP1, SP2 and SP3). The results compared the average of the triplicates reported by the participants. Yellow-highlighted participants are the ones selected for the calculation of the assigned values and underlined the ones that obtained data into ±10% difference to the assigned value for SP2 and SP3.**


The individual performance of each group was further evaluated using z-scores. The results are presented in
Figure 3. All underestimations fall within the acceptable range (lower than -2 z). Additionally, it is noteworthy
that no laboratory exhibits unsatisfactory performances across all three samples; for almost all participants,





while one sample can present poor results, it coexists with two acceptable ones. This has strong implications
for spatial and temporal analyses that are often performed for OP. Again, this calls for attributions that there
are no systematic biases in the analyses. While factors like sample inhomogeneity may be playing a role
(particularly for SP1), some other issues, including variability in the performance of the analysis, may have an
impact. Hence, participants exhibiting significant deviations (|z-scores| > 2) for some of their results should
thoroughly examine their procedures and possibly implement appropriate corrective actions to avoid similar
outcomes in future ILCs.
However, half of the participants achieved results within the acceptable limits of this test. Despite disparities,
these findings are really promising, especially considering that this is the first intercomparison of its kind. For
instance, such results are in the same range to those obtained for some of the first ILCs for PAHs (Grandesso et
al., 2012; Verlhac et al., 2014).

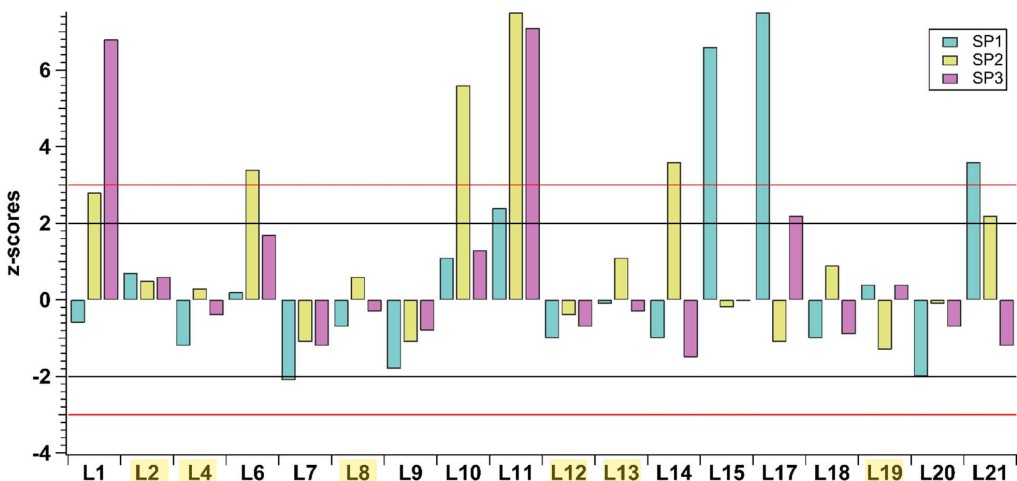


**Figure 3. Z-scores were calculated to evaluate each participant's performance in the interlaboratory comparison for each sample tested. Yellow-highlighted participants are the ones selected for the calculation of the assigned values. Black and red horizontal lines indicate boundaries for triggering an action signal a described in section 2.7.1.**

To gain more knowledge about the factors causing the variability of the results, we first tried to perform a
cluster analysis using the Ward method. This grouped the participants into four clusters (see Figure S8), with
the main cluster (in yellow) including the 10 participants encompassing mainly the ones with satisfactory Z
scores. The clustering seemed independent of the instrument used and/or the time taken between the sample
delivery and analysis (i.e. near the delivery time or later in February).





In a second step, a multiple linear regression model was run to evaluate the associations of the results obtained
for the 3 samples, SP1, SP2 and SP3, considering a range of parameters, including the instrument used and the
delivery and analysis time (Figure 4, Table 2). The beta values are shown in Figure 4, representing the
association (effects) between the different parameters evaluated and the OP results obtained. In the model, the
reference variables were the RI-URBANS DTT SOP and the results obtained with the plate-reader instrument.
Regarding the instrument performance, the values provided by the cuvette-spectrometer were higher than
those obtained with plate readers in the case of SP1 and SP2 (showing significant overestimation in the case of
SP2 p-values <0.05), while the results for SP3 were quite similar. In the case of LWCC, higher variability is
observed when compared to both cuvette and plate reader for all the samples. SP1 LWCC results presented the
highest variability, and SP2 results were significantly overestimated (at a 95 % confidence level) when
compared with those obtained by the plate reader. The RI Urbans SOP was adapted for plate readers and
cuvettes, in order to perform the measurements in similar conditions of concentrations for the reagents. This
was not the case for the LWCC since we did not have all the necessary information concerning the specific
devices used by participants. Figure 4 suggests that the specific conditions of the reaction are probably
important factors for delivering an accurate value of OP. In Figure S6, we showed that SP1's OP activity
decreased over time during storage, but this ageing effect was not found to be significant in the model for either
delivery or analysis time. The storage effect remained consistent for SP2 and SP3, as there was a clear
association between OP and delivery or analysis time (Figure 4).

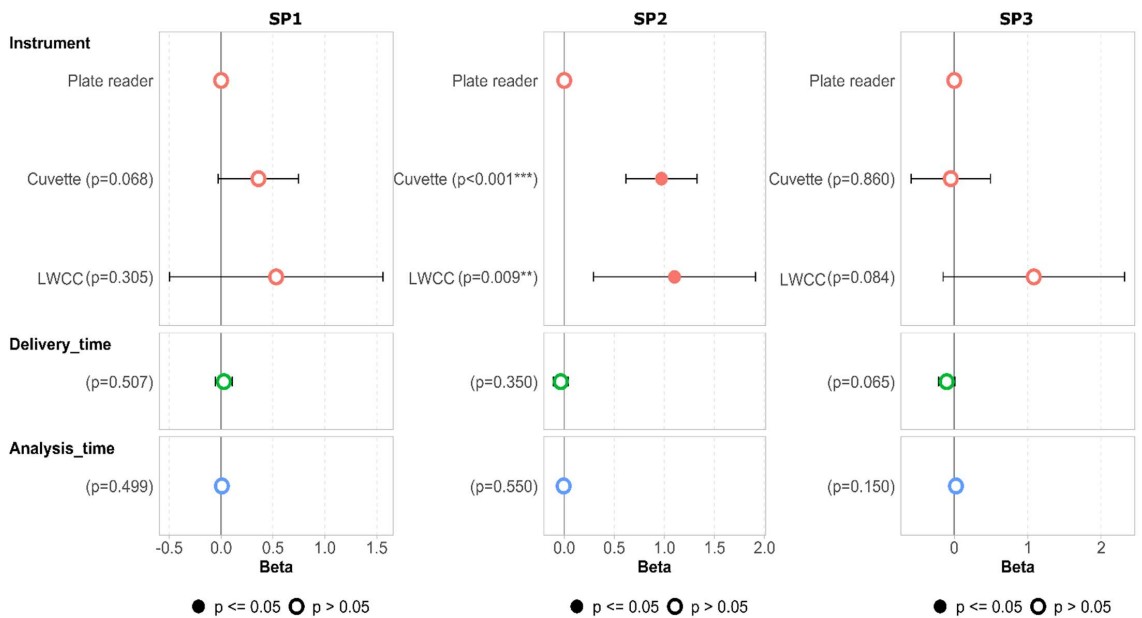






**Figure 4. Associations (beta in nmol min$^{-1}$ μg$^{-1}$) between OP DTT values for SP1, SP2 and SP3 using the RI-URBANS protocol and technical parameters, including the instrument used and the delivery and analysis time obtained by applying an adjusted multiple linear regression model. Full-colour dots represent the results with p-values <=0.05, and white dots represent the results with p-values >0.05.**

A ranking of the samples is also proposed to evaluate the OP activity of the samples tested in this ILC and its relative variability within the participants (only considering the results obtained with the simplified RI-URBANS protocol). For this purpose, SP1 was arbitrarily selected as the one with the highest OP activity with an assigned value of 100, and SP2 and SP3 were evaluated in function of SP1. Figure S12 shows the results obtained for the relative ranking of the samples. It can be noted that most of the participants presented similar relative variability with SP1 > SP2 > SP3. Some exceptions were observed for L1, L17 and L19, which obtained higher ranking for SP3 than SP2. A higher variability in the relative activity is obtained for SP2 than for SP3. Within the participants showing a higher relative ranking for SP2 (higher than 50% compared to SP1), most of them used either cuvette-type or LWCC instruments (except L20), suggesting some overestimation in the results using these instruments. Overall, this similar ranking for the samples achieved by most of the groups is noticeable and very encouraging. In fact, most of the data treatment performed on OP with atmospheric variables or health data relies on associations and regressions where the relative variability of a time series is of utmost importance, more than the absolute value.

**3.5 Comparison with other OP tests provided**

Participants were also invited to report results obtained using other OP assays. Since not all participants submitted results from equivalent "home OP" tests, we exclusively focus on the outcomes obtained through the "DTT-home" protocols involving 13 participants (Table S3, Figure S9). It is important to note that DTT-RI-URBANS protocol was simplified and does not include EDTA, Tris-HCL, TCA neither Chelex®, whereas DTT-"home" protocols are diverse and should exhibit at least one up to 3 of the last mentioned compounds at different step of the reaction; however, this is challenging to evaluate because not all groups have submitted the protocols related to their "home" results.

We have first evaluated the COV individually from the results obtained with protocols (RI-URBANS SOP versus all DTT-"home") for each test sample. Figure 5 shows that lower COVs are generally observed in the performance of the DTT-home protocols (more details can be found in Table S3). However, six out of 12 participants presented similar COVs (within 20%) for the two protocols. These results could indicate that the use of a simplified OP protocol needs some extent of training and guidance before its application. In addition, some of the participants presented higher COV values (L1 and L13 for DTT-home) when using the LWCC instrument. The lack of a simplified protocol for this instrument did not seem to be a major issue, as the application of the DTT-home protocols was also associated with high COV.




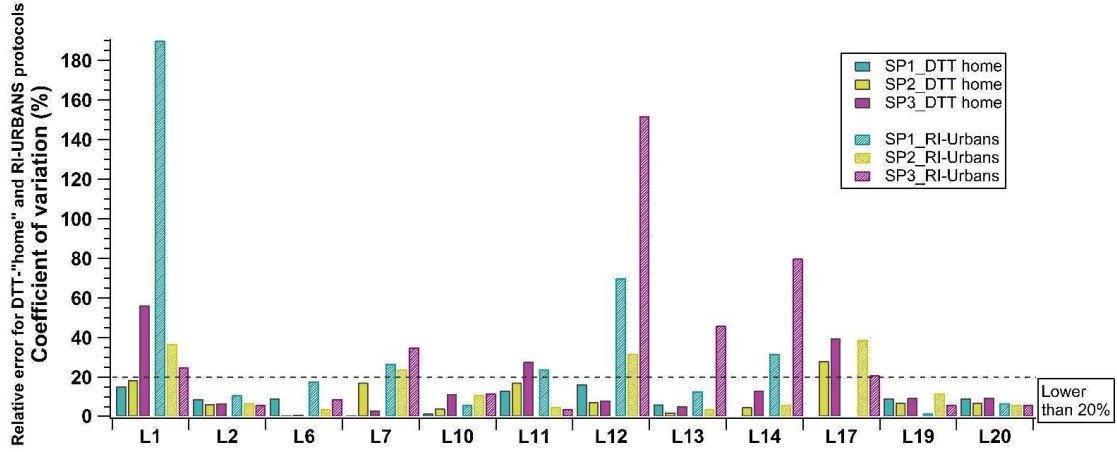


**Figure 5: Coefficients of variation of each participant for the three samples tested from triplicates using both the RI-URBANS DTT and the DTT-"home" protocols. The average and standard deviation of each participant are also detailed in Table S3.**


Another multiple linear regression model was run to evaluate the main differences in the results obtained
between RI Urbans SOP and DTT-"home" protocols (Figure 6). For SP1, a significant overestimation was
observed for the DTT-home protocol and the opposite was observed for SP3. In the case of SP2, there was no
statistically significant difference between both protocols. More details on the concentration of DTT for each
DTT method could provide more insight into this trend. Regarding the instrument performance, the LWCC
presents poorer results compared to the cuvette and plate reader for all the samples, which is opposite to the
trend observed with the results of the RI-URBANS protocol only (Figure 4). However, the results are
significantly underestimated for SP3 only ($p < 0.05$). For the cuvette-based measurements, the results are higher
than those obtained with plate readers in the case of SP1 and SP2 (significantly overestimated for SP2) and
similar for SP3, which is in line with the direction observed with the RI-URBANS protocol only. The delivery
and analysis time show a statistically significant lack of effect for SP2 (analysis time) and an underestimation
for SP3 (delivery time) but nothing significant for SP1, although it had undergone ageing in the tests of the ILC
organiser. Since the effects are very small compared to the effects of the protocol, or the instrument, these two
variables (delivery and analysis time) may cause a greater impact on DTT values when protocols are
harmonised, but not to date.
The results obtained by the participants (the z-scores evaluation) were also added to the former model to
evaluate the effect on OP values while adjusting the protocol variables (Figure S10). The results show a



significant OP overestimation of all the samples for the labs with poor performances in SP2 and SP3 samples,
and also a significant underestimation in the OP value obtained for SP2 for the group with intermediate
performance.
Hierarchical cluster analysis was conducted, incorporating both the DTT "RI-Urbans" and DTT "home"
outcomes (Figure S11). Because some participants did not implement a DTT "home" protocol, the cluster
analysis involved a reduced set of OP values. The results reveal the presence of four primary clusters, with the
predominant cluster encompassing most participants (8 out of 12 for the DTT- "home"). The participants within
the green main cluster largely align with the results derived from the DTT "RI-Urbans" outcomes, encompassing
the groups in the two primary clusters (Figure S11). This assessment illustrates some consistency of results
obtained across various OP DTT protocols. Some of the participants with more reliable results for the RI-
URBANS DTT SOP maintain their consistency regardless of the protocol used. However, some of those that did
not show an acceptable performance for the simplified protocol (i.e. L1 and L10) presented a better
performance for the DTT-"home", and the opposite was observed for the L19 (almost for SP1).

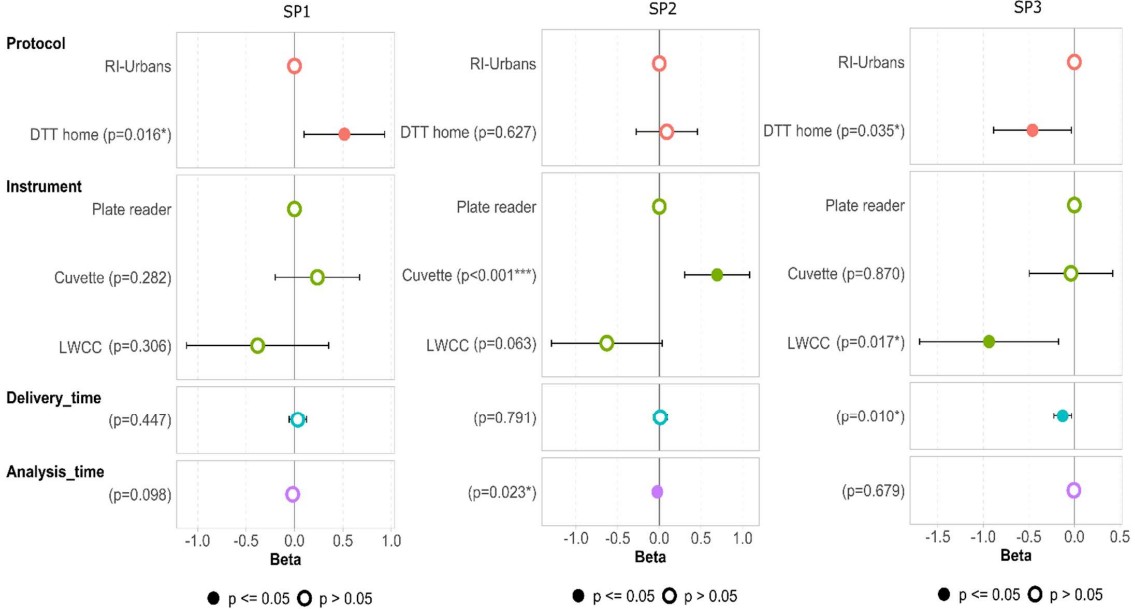


**Figure 6. Associations (beta in nmol min$^{-1}$ µg$^{-1}$) between OP DTT values obtained for SP1, SP2 and SP3**
**and the different parameters of the ILC, including the DTT protocol, the instrument used and the**
**delivery and analysis time obtained by applying a multiple linear model. Full-colour dots represent the**
**results with p-values <=0.05, and white dots represent the results with p-values >0.05.**






Finally, to assess the performance of the participants in the DTT-"home" protocols, a comparable approach to
the simplified RI_URBANS SOP was employed for those participants who supplied OP results. The z-scores were
computed using the assigned values of each sample (SP1-SP3), obtained with RI-URBANS SOP application.
Figure 7 illustrates the z-scores of the OP results obtained through the application of the DTT-"home" protocols,
revealing a significant variation in the outcomes. Only five participants managed to produce satisfactory results
for all the tested samples. Despite the fact that the COV of the participants using DTT-"home" protocols showed
an improvement over the results of the simplified DTT SOP (Figure 5), the outcomes are still distant from the
consensus values of the samples obtained in this exercise. The results indicate a high degree of variability in
the OP activity using "home" OP methodologies, underscoring the pressing requirement for standardized
methods and harmonised protocols to ensure more reliable OP research.

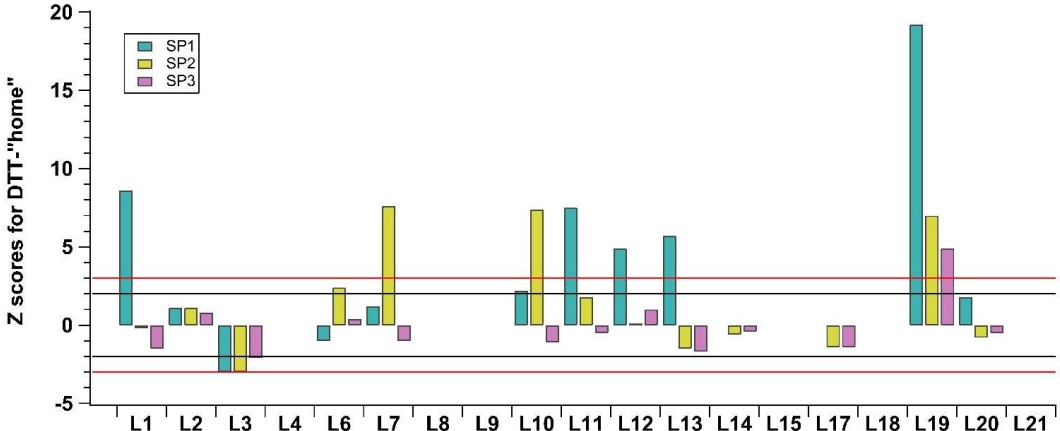


**Figure 7. Z-scores were calculated for the DTT-"home" protocol results to evaluate each participant's**
**performance in reference to the RI_URBANS assigned (consensus) values, for each sample tested. Black**
**and red horizontal lines indicate boundaries for triggering an action signal a described in section 2.7.1.**

**4.  Strengths and limitations of this first intercomparison**
The greatest strength of this ILC was the high number of participants (20) enhancing the comprehensiveness
and diversity of the study and allowing for a broader range of perspectives and expertise. This also allowed
comprehensive collaborative discussions during the preparation phase, promoting knowledge exchange and
consensus-building, and contributing to a more robust ILC design. These all show a willingness from the groups



to be actively part of the development of the intercomparison and to pursue a harmonisation on the OP
measurements.
The development of the first simplified OP DTT protocol (available in SI-1) also consolidates the experimental
experience of the participants, fostering methodological consistency across different research groups as a first
step toward method harmonisation. Finally, the collaboration with the JRC, an independent organisation for the
assessment of results, adds credibility and objectivity to the study, ensuring that findings are impartially
evaluated.
The sharing of liquid samples in this comparison comes with both advantages and limitations. On one hand, it
eliminates biases associated with extraction methods and solvent purity. However, some samples exhibited
signs of ageing during the interlaboratory comparison duration (though this was not identified as a critical
parameter when identifying the main causes of variability). In addition, this approach introduced certain
challenges with some of the "home" OP protocols which were designed originally to be used with solid samples.
Finally, all liquid extracts should be provided with a similar PM concentration to limit known nonlinearities and
to avoid potential saturation issues, as can be the case of LWCC instruments.  Next, ICL should, in the future,
include the whole chain assessment, including the extraction step.
The testing of three samples with different patterns makes it difficult to draw unilateral conclusions. A larger
sample size could support a more robust statistical analysis of the results, particularly for the factors
determining OP variability. Moreover, the inclusion of samples that are readily accessible worldwide, such as
standard reference materials, could facilitate the future adaptation of a simplified OP DTT protocol while
allowing comparison with an assigned (consensus) value.
The sole focus on the DTT method for the ILC could limit the broader evaluation of OP. The addition of other
OP assays, such as those included in previous inter-comparison studies (Ayres et al., 2008; Calas et al., 2018;
Shahpoury et al., 2022) could provide a more comprehensive understanding of the performances of the
different groups involved in the OP domain of research. There are contrasting reports about the relative
sensitivity of DTT assay to various organic and inorganic PM components, with some studies showing higher
reactivity towards the organic fraction. Therefore, additional consensus studies would be needed to assess this
aspect and the comparability of DTT to other OP metrics that rely on proper lung anti-oxidants and could be
considered more physiologically relevant. Such studies could support the identification of chemical species
which should be prioritized for future air quality management programs.
In conclusion, while this ILC of OP has highlighted considerable variability in the performance of the assay
between groups, it has notable strengths and provides a starting point towards the harmonisation of OP
measurements.

**5.  Recommendations for standardisation of OP protocols**





Based on the findings of this ILC and also on general literature about OP, some recommendations for further
standardising OP DTT measurements are proposed (Table 1). These include guidelines for sample and
laboratory conditions, instrument type and calibration and the reporting of results. Additionally, a reference
material (1,4-naphthoquinone, copper or other solution at a known concentration) should be proposed to
facilitate future ILCs. Additionally, since some differences were observed in the results obtained from the use
of the simplified SOP compared with the "home-developed" DTT protocols, harmonisation of procedures is
needed to ensure data comparability. We describe below the crucial parameters that need to be considered in
the move toward greater harmonisation of OP methodologies.
*OP assay selection*
• To date, it remains unclear which oxidative potential (OP) assay is most effective at predicting health
outcomes related to oxidative stress. This uncertainty arises because different assays yield markedly different
results for the same particulate matter (PM) samples. Additionally, even OP values from the same assay are
often linked to various PM components and sources, depending on the different studies (He and Zhang, 2022).
Thus, based on current knowledge and epidemiological evidence, two complementary OP assays (a thiol-
based probe (OPDTT or OPGSH) and another one (among OPOH, OPAA, or else)) are recommended to provide
a better picture of the potential oxidising damages from PM compounds and to strengthen the power of
epidemiological studies. These aspects were previously discussed in a recent work integrating five different
OP assays (Dominutti et al., 2023). Finally, the final choice of the best OP test (or combination) must be based
on epidemiological evidence, which has begun only recently and needs more hindsight to be determined.
*Sampling*
• OP can be analysed in filter samples conventionally collected for air quality monitoring using small
portions of these if adequately preserved (frozen). Pre-burn quartz filters or Teflon filters are appropriate
and blank filters must be measured to remove the background induced by the matrix of the material. A
previous study had shown no differences in the OPDTT values observed using Quartz or Teflon filters
(Frezzini et al., 2022). However, it should be further evaluated when other OP assays are considered.
*Sample storage*
• Previous studies that evaluated the effect of storage time and conditions did not show a substantial effect
on the OP DTT results (Frezzini et al., 2022). However, we recommend that PM samples should be
immediately transported to the lab after sampling. The filters must be kept cold after sampling (at 4°C if the
OP analysis is done within a few days after collection or -20°C if the analysis is delayed).
• The lifetime of the ROS may be very short, and measurements of OP on PM-extracted filters are likely
affected both by the age of the samples, how they have been sampled and stored, and the nature of the
extraction methodology. How all of these processes impact on the ageing of samples and the ultimate





quantification of OP needs to be addressed. Ageing studies should be performed for each OP assay in the long
term to define the maximum storage time of aerosol filters at low vs ambient temperature conditions.
Some OP components might be so short-lived that only online techniques are warranted.
*Laboratory conditions*
• OP assays are "trace" detection assays that require clean ambient conditions and high-quality reagents
free of metal contamination. Considerations should be given to the use of certified clean rooms or proper
laminar flow bench stations to prevent contamination of the samples.
• Use of clean material: vials, cones, and spatulas have to be washed before use (5% HNO3 bath to remove
metals and rinse three times in ultra-pure water before drying in laminar flow).
• Control laboratory temperatures and light exposure by using dark polypropylene tubes at least for
reactants
*Extraction step*
• The extraction step may be highly variable according to the procedures used, and several parameters are
known to impact OP results, such as the choice of the solvent, the concentration of buffer, the way of agitation,
and the quantification of the final extracted mass. Notably, the ultrasonication of PM samples in aqueous
solutions generates ROS (Miljevic et al., 2014) and it could introduce artefacts in OP measurement. This effect
was also observed in the work of Frezzini et al. (2022), where different extraction methods were evaluated,
with ultrasonic baths overestimating the results observed.
The effect of the solvent used was not evaluated in this ILC exercise. However, we recommend the use of
ultrapure water or simulated lining fluid for the sample extraction. Future ILC exercises should include the
evaluation on the extraction conditions, including solvent use and methods.
*Reaction step*
• Several aspects in the reaction process affect the OP value, like the initial concentration of reactants (since
the DTT test is mass-dependant (Charrier et al., 2016)), ratio of reactant/sample, time of reaction (some
compounds present a non-linear reaction over time), the temperature of the reaction (which should be
standardised to 37°C), agitation (mixing samples) and the type of measurements (kinetic or end-point value),
etc.
• Current literature mainly addresses extraction or reaction parameters separately. We advise that the
whole chain factors should be evaluated together to quantify their relative impact on the results.
*Development of a reference material with a certified "OP value."*
The setup of reference material or in-house standard solutions (in collaboration with reference institutions
JRC or NIST, for instance) with a known OP value could help laboratories test and train themselves on the OP
protocol before testing the unknown ILC samples. This is something to be developed and tested in future ILCs.



*Instrument calibration*
• Investigate the optimal frequency for the calibration of spectrophotometers for such assays.
*Report of results/units*
The calculation of OP DTT activity during this ILC involved a conversion using a calibration curve. Since the
OP activity measures the rate of a chemical reaction and not a concentration, for future comparison exercises,
the possibility of exploring alternative methods for OP calculation should be tested. To date, results are mass
normalised in nmolAnti-oxidant min$^{-1}$ µg$^{-1}$, or volume normalised in nmolAnti-oxidant min$^{-1}$ m$^{-3}$. The OP per
µg refers to the reactivity of one µg of the tested PM, whereas the OP per m$^3$ refers to the exposure of one m$^3$
of inhaled air.
Table1. Summary of recommendations for future OP measurements on filters

| Condition or step | Recommendations |
|---|---|
| OP assay selection | • Two complementary OP assays (a thiol-based probe (OPDTT or OPGSH) and another one (among OPOH, OPAA, or else)) are recommended to provide a better picture of the potential oxidising damages from PM compounds |
| Sampling | • Pre-burn quartz filters or Teflon filters are appropriate and blank filters must be measured to remove the background induced by the matrix of the material. |
| Samples storage | • The PM filters must be kept cold after sampling (at 4°C if the OP analysis is done within a few days after collection or -18°C or -20°C if the analysis is delayed). |
| Laboratory conditions | • Clean conditions (including certified clean rooms or proper laminar flow bench stations)<br>• High-quality reagents free of metal contamination<br>• Use of clean material, which must be washed before use (5% HNO$_3$ bath to remove metals, rinse three times in ultra-pure water, and dry in laminar flow bench stations).<br>• Control laboratory temperatures and light exposure by using dark polypropylene tubes |
| Reaction step | • Several aspects in the reaction process affect the OP value, and to minimise their impact, standard conditions should be fixed as the initial concentration of reactants, ratio of reactant/sample, time of reaction, the temperature of the reaction (37°C), agitation (mixing samples) and the type of measurements (kinetic or end-point value), etc. |
| Instrument calibration | • Investigate the optimal frequency for the calibration of spectrophotometers for such assays. |


**6. Conclusions**
This study represents an innovative effort as the first interlaboratory OP exercise specifically aimed at
harmonising this OP assay. This exercise provides the very first roadmap for refining interlaboratory



comparisons of OP, fostering greater confidence in the reliability of OP data and encouraging the scientific
community to advance towards global OP harmonisation.
This first exercise focused on OP DTT, as it is widely used within the scientific community and has already
shown positive associations with health outcomes (Bates et al., 2015; Borlaza et al., 2022; Dabass et al., 2018;
Donaldson et al., 2001; Gao et al., 2020; Marsal et al., 2023; Weichenthal et al., 2016b, a, c). Even if there are
several crucial points to be evaluated and harmonised in the whole chain of the determination of OP (sampling
methods, sample storage, extraction conditions and methods) as well as the use of different OP assays, this first
ILC engaging several research laboratories pay the way for future developments towards the standardisation
of OP methods. Despite the need to evaluate and harmonize several crucial aspects throughout the entire
process of OP measurements (such as sampling methods, sample storage, extraction conditions and methods,
and the use of different OP assays), this initial ILC, which involves multiple research laboratories, paves the way
for future advancements in the standardization of OP methods.
Our findings emphasise both the strengths and challenges associated with the use of the current OP DTT assay
for driving a measurement of PM OP. Overall, half of the participants achieved results falling within a
satisfactory range of z-scores for this test. The participating group performance levels are comparable to those
observed in initial ILCs for PAHs in the 2010s (Grandesso et al., 2012; Verlhac et al., 2014). While notable
agreement was observed in certain samples and between several groups, discrepancies and variability were
also identified, emphasizing the need for harmonisation in the procedures and conditions. A number of factors
may contribute to the underperformance observed in certain samples and participants. The main reasons are
not clear, but the analysis conditions in the participating laboratories and the lack of experience in this type of
metrological exercise are possible causes. Standardisation of protocols and harmonisation of procedures
emerged as critical components to ensure the accuracy and comparability of OP data across laboratories. This
collaborative approach fosters a more robust OP science, facilitates data exchange and integration, and will
ultimately contribute to a better understanding of the health impacts associated with PM exposure, allowing
for more accurate exposure assessments and regulatory decisions.

**Acknowledgements**
The present work was supported by the European Union's Horizon 2020 research and innovation program
under grant agreement 101036245 (RI-Urbans), including the Post-doc grant of Pamela Dominutti, and by a
University Grenoble Alpes grant ACME IDEX (ANR-15-IDEX-02). Chemical analysis of the Air-O-Sol facility at
IGE was made possible with the funding of some of the equipment by Labex OSUG@2020 (ANR10 LABX56). DP,
AB and NM acknowledge support by the project REGENERATE funded by the Hellenic Foundation for Research
and Innovation (HFRI) through the General Secretariat for Research and Technology (GSRT), Project No. 3232.
HC and PM acknowledge support by project No. 24-10051S funded by the Czech Science Foundation. The





analyses in Serbia were supported by the EC Horizon Europe research and innovation program under grant
agreement GA 101060170 (WeBaSOOP). Analyses at ISAC-CNR were supported by the project PER-ACTRIS-IT,
funded by the Ministry of Education and Research (MIUR), Action II.1 of PON Research and Innovation 2014-
2020. This study was also supported by the PROCOPE mobility program (0185-DEU-23-0008 LG1) for grant
0185-DEU-23-0008 LG1 to Eduardo Souza (ACD, TROPOS). PS acknowledges support from the Hazardous Air
Pollutant Laboratory of Environment and Climate Change Canada. BU and MK acknowledge funding by the
Swiss National Science Foundation, grant 200021_192192/1. IM received funding from the National Institute
for Health Research (NIHR) Health Protection Research Unit in Environmental Exposures and Health, a
partnership between the UK Health Security Agency and Imperial College, with IM receiving additional funding
from the MRC Centre for Environment and Health, which is funded by the Medical Research Council
(MR/S0196669/1, 2019–2024).
We also acknowledge Mark Diks (TNO), Alexandre Barth and Julian Resch (UNIBAS), Laurent Alleman (IMT-
Nord) and Stephan Houdier (IGE) for providing help on the deployment of this ILC.
**Data availability**
All OP data is published in the SI.
**Author contributions**
GU designed the study and supervised the OP tests for the development of the simplified DTT RI-URBANS
protocol. CR, TM, RE and PAD tested the different steps and prepared the samples and logistics of the
intercomparison exercise. GH, R.M.H, A.N, I. M., K. B, N.M and G.U. evaluated and decided the samples to be
compared and the protocol content. JPP and FC performed the data analysis and evaluated the performance of
each group. PAD and AM developed the multiple regression model. PAD processed the data of the study and
wrote the paper together with JLJ and GU before a first review by the RI-Urbans members. All authors
participated in the interlaboratory comparison. All authors reviewed and edited the manuscript.
**Competing interests**
At least one of the (co)-authors is a member of the editorial board of Atmospheric Measurement Techniques.

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
