# Peer review of "An interlaboratory comparison to quantify oxidative potential measurement in aerosol particles: challenges and recommendations for harmonisation"

_Atmospheric Measurement Techniques, 2024_

## Referee Comment (RC1)

Author: Dominutti PA… Uzu G

Title: **An interlaboratory comparison to quantify oxidative potential measurement in aerosol particles: challenges and recommendations for harmonisation**

**General comments**

The inter-laboratory comparison (ILC) involves 18 laboratories performing the DTT-based OP assay on four samples (SP1-SP4). Such an ILC is very much needed to achieve the aim of harmonizing this type of OP across different labs and ultimately to ensure comparability of monitoring OP as a proxy measure of PM toxicity over a wide spatial coverage and a longer temporal span.  I have a few major comments, which are detailed below, regarding the analysis of measurement uncertainty sources.

 **Major comments**

(1) Reading the SOP described in S1-1, I deduce the equation to calculate the DTT consumption rate in nmol min$^{-1}$ for a sample is:

$$\Delta DTT_t \ (nmol \ min^{-1}) = \frac{V}{\Delta t}\left(DTT_o - \frac{A_t - A_{int}}{k}\right) - \Delta DTT_{a.o.r} \qquad \text{Eq (1)}$$

Where V is the volume of the DTT solution for which absorbance measurement is taken; $\Delta t$ is the incubation time (e.g., 10, 20, 30 min),$DTT_o$ is the initial DTTT concentration used in the assay;  $A_t$ is final absorbance reading for T= $t$ min incubation experiment (from step 16); $A_{int}$ is the intrinsic absorption of each sample (from step 7), $k$ is the calibration slope of absorbance vs concentration of DTT, and $\Delta DTT_{a.o.r}$ is the inherent DTT auto-oxidation rate (slope of Control$_{ox}$ sample).

(Note: It is unclear about $DTT_o$ (the initial DTTT concentration used in the assay) is determined. Is it calculated from the known concentration (0.25mM) and the dilution factor (300/50 =6)? Or it is calculated from Absorbance measurement at incubation time 0 min?)

With the calculation equation established, a few questions ensue:

First, the calculation equation (1) needs to be provided in the SOP to avoid any second guessing on readers' or users' part. Second, if Eq (1) is the correct interpretation of the DTT oxidation rate calculation, then we can see $A_t$,  $A_{int}$, $k$, and $\Delta DTT_{a.o.r}$  all contribute to the measurement uncertainty to $\Delta DTT_t$. Their measurement values and uncertainties (COV) from participating labs need to be presented in the paper so that we can understand which measurement step/variable contributed most to the overall uncertainty of $\Delta DTT_t$, especially for those labs that produced results of larger COV for the ILC samples (e.g., L1, L12, L13, L14, L15, and L21 shown in Figure 1).

(2) Lines284-286: "SP4: a sample extracted from a blank/clean quartz filter was sent to the participants, but it was not included in the evaluation since the measured values

were close to the instrument limits of detection for most participants." It is misleading when stating "close to the instrument limits of detection. If eq. (1) shown in the preceding comment is correct, then SP4's $\Delta DTT$ being not detected is actually the first term in eq (1) being indistinguishable to $\Delta DTT_{a.o.r.}$. Then, it is relevant to characterize the COV of $\Delta DTT_{a.o.r.}$. $\Delta DTT_{a.o.r.}$ is likely lab-specific, as ultrapure water and reagents (e.g., DTT, phosphate buffer, DTNB, etc) were prepared by individual participating labs. The lab cleanliness conditions and source and storage history of reagents could all affect ΔDTTa.o.r. and its variability.

In any case, the ILC results for SP4, as well as $\Delta DTT_{a.o.r.}$, needs to be discussed. Related to this, the limits of detection as related to determination of $\Delta DTT$ merits clarification.

(3): Selection of the four ILC samples: please elaborate reasons for their selection. Why a copper standard solution is not considered as an ILC sample, considering the strong response of OP DTT to Cu?

(4) Shown by Figure 3, all underestimations are lower than -2 z, however, 8 labs had one or more z-scores exceeding +2 z. This implies propensity for these labs to get overestimations. Do these labs tend to have higher $\Delta DTT_{a.o.r.}$?

**Minor comments**

- Line 313: Here the authors state that a total of 20 research groups participated in the ILS, but Figures 1, 3, 7, and S7 show laboratory labels up to "L21". Please explain the discrepancy.
- S1-1: "weight" is mistakenly used throughout the description of Method 1. Please replace with "weigh".
- line 264 and S1-1, line 52: why the experiment temperature was 37.4oC, not 37oC?
- Please provide instruction on preparing particulate matter suspension solutions in the simplified DTT RI-URBANS SOP.
- Line 288: "SP3 were powders": unclear how could SP3 (originated from urban PM) could be in powder form?
- Line390: insert a comma between "solutions" and "liquid".
- Define COV at its first appearance (line 337-338) instead of later on (line 403)
- Figure 5: suggest using different legends for DTT-"home " and R1-URBANS protocols for easy differentiation. For example, hatched lines could be included for DTT-"home" data.
- Line 712: pay the way → pave the way
- Line 713-716: Text here is largely redundant, as it repeats much of the preceding sentence.

---

## Author Comment (AC1)

**AMT review: amt-2024-107, An interlaboratory comparison to quantify oxidative potential measurement in aerosol particles: challenges and recommendations for harmonisation**

Reviewer #1

The strength of this manuscript lies in its cross-laboratory comparison of the DTT assay for evaluating the oxidative potential (OP) of particulate matter. The study's objective—assessing the assay's reliability by involving a relatively large number of laboratories and identifying uncertainties—was clear and well-intentioned. However, I found the analysis of the results lacking in solid statistical foundations. Some data treatments appear arbitrary and lack justification, such as the selection of "assigned values" for bias evaluation, which raises questions about key interpretations. The manuscript can provide clearer statistical inferences, and some results, such as the COV and z-score, appear to overlap statistically, adding redundancy rather than clarity. Citations for some statements are vague, imprecise, or insufficient. While the manuscript presents common statistical values from various labs, it is unclear how these observations provide practically beneficial insights into the variation in outcomes and their underlying causes. Some specific comments are given below.

We thank the reviewer for their feedback, though we note that not all comments were equally constructive. Nevertheless, we have addressed the various issues and questions raised in the review.

We respectfully disagree with the assertion that the analysis of the Coefficient of Variation (COV) and z-scores is redundant. The COV assesses the intra-laboratory variability of internal triplicates, while the z-scores evaluate the average values of each sample and participant in relation to the assigned (consensus) values. Although some participants demonstrated precise and consistent measurements, their results did not align with the consensus values.

L99: I cannot reference the citation "WHO 2017" in the reference list.

We have added and updated this reference by WHO 2024.

L 102-103: I find it difficult to precisely support the claims from the references- "Huang et al., 2022; Nicholson et al., 2022; Wilker et al.,103 2023; Zare Sakhvidi et al., 2022 ". I did not find evidence from these studies to link PM to mortality. I also recommend increasing the citations' precision- e.g., cite the exact reference to support the oxidative stress and inflammation in the statement.

Thank you for this comment. We now realize that the transition was unclear, as the references were intended to correspond with the different alterations mentioned at the end of the sentence. We have revised the manuscript to correct this as follows:

*The casual mechanisms underpinning these adverse associations are diverse, with oxidative stress and inflammation (Leni et al., 2020; Li et al., 2003), genomic alterations (Huang et al., 2022),damage to the nervous system function (Wilker et al., 2023), and epigenetic alterations, cognitive decline (Nicholson et al., 2022; Zare Sakhvidi et al., 2022), among others, all cited as potential contributing pathways.*

L 106: I don't find this reference supporting the claimed "unifying mechanism" from "Li et al., 2018".

The reference is Li et al. (2008). In this study, the authors conducted a comprehensive review of the mechanisms underlying particulate matter (PM)-induced oxidative stress and the various cellular response pathways influenced by different levels of oxidative stress. They demonstrated that these oxidative stress pathways can ultimately lead to biological damage at the cellular level.

It could be possible that "unifying" is not the best word choice here, we have then corrected the text as follows:

*Across these broad domains, the capacity of particles and particle-derived chemicals to induce damaging biological oxidations appears to be primarily linked to oxidative stress. This occurs not only through the direct introduction of pro-oxidants and stable free radicals into the body, but also through secondary radical and oxidant generation via altered metabolism and inflammation induction (Li et al., 2008).*

L 333: How do the authors determine the number of 20 here?

This sentence was removed and corrected.

L342 - 343: It is unclear to me what "values obtained for the dataset" that is suggested to jeopardize the use of the algorithm. L348-351: The authors appear to arbitrarily toss the high-variance outcome. How to justify this without compromising the statistical significance?

The selection of the "consensus among expert laboratories" approach is justified by the lowest uncertainty associated with the assigned values obtained through this method, compared to the two others. We excluded the option of relying on a single laboratory's results, as this would require that the laboratory demonstrate the accuracy of its measurements using reference materials—materials that are currently unavailable commercially and not existing for OP.

We have revised this section to clarify the rationale behind our method selection as follows:

*…The standard uncertainty of the assigned value shall be as small as possible to minimize the risk that participants will receive underperformance signals because of inaccuracy in the determination of the assigned value. Three methods including the use of (i) a simple mean and standard deviation including all participants, (ii) the Q/Hampel test, and (iii) the consensus value and standard deviation from expert laboratories were compared…*

L369-373: Describing more details of the statistical method would be beneficial.

This statistical method integrates participants' performance, represented by z-scores, along with data on the instruments used and the different time points of measurement. It offers an alternative perspective on the under- and overestimation of results, seeking to identify potential associations with underlying causes.

**Reviewer #2**

General comments

The inter-laboratory comparison (ILC) involves 18 laboratories performing the DTT-based OP assay on four samples (SP1-SP4). Such an ILC is very much needed to achieve the aim of harmonizing this type of OP across different labs and ultimately to ensure comparability of monitoring OP as a proxy measure of PM toxicity over a wide spatial coverage and a longer temporal span. I have a few major comments, which are detailed below, regarding the analysis of measurement uncertainty sources.

We thank the reviewer for this detailed evaluation of our work. We have addressed the comments and questions below.

Major comments

(1) Reading the SOP described in S1-1, I deduce the equation to calculate the DTT consumption rate in nmol min-1 for a sample is:

$$\Delta DTT_t \ (nmol \ min^{-1}) = \frac{V}{\Delta t}\left(DTT_o - \frac{A_t - A_{int}}{k}\right) - \Delta DTT_{a.o.r} \qquad \text{Eq (1)}$$

Where V is the volume of the DTT solution for which absorbance measurement is taken; $\Delta t$ is the incubation time (e.g., 10, 20, 30 min),DTTo is the initial DTTT concentration used in the assay; At is final absorbance reading for T= t min incubation experiment (from step 16); Aint is the intrinsic absorption of each sample (from step 7), k is the calibration slope of absorbance vs concentration of DTT, and $\Delta DTT$a.o.r is the inherent DTT auto-oxidation rate (slope of Controlox sample).

(Note: It is unclear about DTTo (the initial DTTT concentration used in the assay) is determined. Is it calculated from the known concentration (0.25mM) and the dilution factor (300/50 =6)? Or it is calculated from Absorbance measurement at incubation time 0 min?)

The equation to calculate the DTT consumption rate (nmol min$^{-1}$) is based on the absorbance measurement from the one obtained at incubation time 0 min, on the calibration slope, and on the DTT auto-oxidation rate, as follows:

$$\Delta DTT_t \ (nmol \ min^{-1}) = \frac{V}{\Delta t}\left(\frac{(A_t - A_{t0})}{k}\right) - \Delta DTT_{a.o.r}$$

Where V is the volume of the DTT solution for which absorbance measurement is taken; $\Delta t$ is the incubation time (e.g., 10, 20, 30 min), At is final absorbance reading for T= t min incubation experiment; At0 is the intrinsic absorption of each sample, k is the calibration slope of absorbance vs concentration of DTT, and $\Delta DTT$a.o.r is the inherent DTT auto-oxidation rate (slope of Controlox sample).

With the calculation equation established, a few questions ensue:

First, the calculation equation (1) needs to be provided in the SOP to avoid any second guessing on readers' or users' part.

The equation was included in the SOP in the supplementary information as requested.

Second, if Eq (1) is the correct interpretation of the DTT oxidation rate calculation, then we can see At, Aint, k, and $\Delta DTT$a.o.r all contribute to the measurement uncertainty to $\Delta DTT$t. Their measurement values and uncertainties (COV) from participating labs need to be presented in the paper so that we can understand which measurement step/variable contributed most to the overall uncertainty of $\Delta DTT$t, especially for those labs that produced results of larger COV for the ILC samples (e.g., L1, L12, L13, L14, L15, and L21 shown in Figure 1).

We agree with the reviewer that such an evaluation could be valuable in identifying the key factors contributing to the uncertainty for each participant. However, the aim of this paper was to provide a general overview of the main results across the different groups working on OP measurements and crucial stages (transport, type of instrument etc etc), rather than a detailed analysis of individual uncertainties. Nevertheless, we believe the findings of this study can still guide participants in examining their internal errors and identifying potential shortcomings in their results, even though this falls outside the scope of the current work.

2) Lines284-286: "SP4: a sample extracted from a blank/clean quartz filter was sent to the participants, but it was not included in the evaluation since the measured values were close to the instrument limits of detection for most participants." It is misleading when stating "close to the instrument limits of detection. If eq. (1) shown in the preceding comment is correct, then SP4's $\Delta DTT$ being not detected is actually the first term in eq (1) being indistinguishable to $\Delta DTT$a.o.r. Then, it is relevant to characterize the COV of $\Delta DTT$a.o.r.

*ΔDTTa.o.r.* is likely lab-specific, as ultrapure water and reagents (e.g., DTT, phosphate buffer, DTNB, etc) were prepared by individual participating labs. The lab cleanliness conditions and source and storage history of reagents could all affect ΔDTTa.o.r. and its variability.

In any case, the ILC resiults for SP4, as well as *ΔDTTa.o.r.*, needs to be discussed. Related to this, the limits of detection as related to determination of *ΔDTT* merits clarification.

SP4 was not the blank used in the first term of Equation 1, but rather a sample sent to participants for testing, representing the extraction of a non-impacted blank filter. As noted, participants were unaware of the sample contents to prevent any bias in their analyses. Given that many of the results for SP4 were very very low, close to zero, the associated uncertainties were then very high, making it impossible to evaluate SP4 using the same statistical methods as the other samples.

(3): Selection of the four ILC samples: please elaborate reasons for their selection. Why a copper standard solution is not considered as an ILC sample, considering the strong response of OP DTT to Cu?

The core group members agreed upon the selection of samples, with four being defined as appropriate for this first ILC: one reference solution, two ambient samples, and one blank.

Regarding the use of copper solutions, we conducted extensive preliminary tests with CuCl2 prior to the ILC exercise. However, it proved unstable over time in Milli-Q water-based solutions. To prevent this instability, we would have needed to prepare the solution under acidic conditions, which would not accurately reflect physiological conditions.

(4) Shown by Figure 3, all underestimations are lower than -2 z, however, 8 labs had one or more z-scores exceeding +2 z. This implies propensity for these labs to get overestimations. Do these labs tend to have higher *ΔDTTa.o.r.*?

Higher ΔDTTa.o.r values were not consistently observed for all the participants exceeding +2 z-scores. For some cases, the COV for ADTTa.o.r was higher than 20% for L1, L15, and L17, but it was lower than 10% for L6, L10, L14, and L21. Thus, the overestimations observed do not seem to be exclusively related to the instrument or the water quality used for these participants.

**Minor comments**

• Line 313: Here the authors state that a total of 20 research groups participated in the ILS, but Figures 1, 3, 7, and S7 show laboratory labels up to "L21". Please explain the discrepancy.

As noted, participant L5 does not appear in any of the figures or tables. This is because they withdrew after the ILC had already been launched. Since the laboratory numbers had already been assigned by that time, we did not want to reassign them.

• S1-1: "weight" is mistakenly used throughout the description of Method 1. Please replace with "weigh".

This was corrected as suggested.

• line 264 and S1-1, line 52: why the experiment temperature was 37.4oC, not 37oC?

This was selected as the physiological temperature it is at 37.4 °C.

• Please provide instruction on preparing particulate matter suspension solutions in the simplified DTT RI-URBANS SOP.

As outlined in the recommendations and conclusion of this work, several key aspects in the OP determination process—such as sampling methods, sample storage, and extraction conditions—need to be evaluated and harmonized. However, the primary focus of this first ILC was the development of a simplified protocol for the DTT assay, with all PM suspension solutions provided and pre-prepared. More details about recommended extraction procedures will be included in the next ILC for AA and DTT expected for the beginning of 2025.

However, we have now included the reference for the method used for sample extraction in this ILC, which is following the method suggested by Calas et al. (2017), which recommends 75 minutes of vortexing in Milli-Q water solutions.

• Line 288: "SP3 were powders": unclear how could SP3 (originated from urban PM) could be in powder form?

Yes, this was inaccurate and we have changed it in the manuscript. The SP1 was obtained from powder and the SP3 was obtained from a concentrated extract that we have to dilute to obtain the concentration proposed.

• Line390: insert a comma between "solutions" and "liquid".

This was corrected as suggested.

• Define COV at its first appearance (line 337-338) instead of later on (line 403)

The text was changed and now COV is defined in its first appearance.

• Figure 5: suggest using different legends for DTT-"home " and R1-URBANS protocols for easy differentiation. For example, hatched lines could be included for DTT-"home" data.

Thanks for this comment. That was already the case, but it seems that the pattern used was not clearly distinguished. We have changed it for a better visualisation.

• Line 712: pay the way = pave the way

This was corrected as suggested.

• Line 713-716: Text here is largely redundant, as it repeats much of the preceding sentence.

This was corrected as suggested.